# Estimated Prevalence of Tuberculosis in Ruminants from Slaughterhouses in Constantine Province (Northeastern Algeria): A 10-Year Retrospective Survey (2011–2020)

**DOI:** 10.3390/life13030817

**Published:** 2023-03-17

**Authors:** Nadir Boudjlal Dergal, Mohamed Ghermi, Kálmán Imre, Adriana Morar, Ulaș Acaroz, Damla Arslan-Acaroz, Viorel Herman, Abdelhanine Ayad

**Affiliations:** 1Laboratory of Biotechnology for Food Security and Energetic, Department of Biotechnology, Faculty of Natural and Life Sciences, University of Oran 1, Ahmed Ben Bella, Oran 31000, Algeria; 2Laboratory of Microorganisms Biology and Biotechnology, Department of Biotechnology, Faculty of Natural and Life Sciences, University of Oran 1, Ahmed Ben Bella, Oran 31000, Algeria; 3Department of Animal Production and Veterinary Public Health, Faculty of Veterinary Medicine, University of Life Sciences “King Mihai I” from Timișoara, 300645 Timisoara, Romania; 4ACR Bio Food and Biochemistry Research and Development, Afyonkarahisar 03200, Turkey; 5Department of Food Hygiene and Technology, Faculty of Veterinary Medicine, Afyon Kocatepe University, Afyonkarahisar 03200, Turkey; 6Department of Food Hygiene and Technology, Faculty of Veterinary Medicine, Kyrgyz-Turkish Manas University, Bishkek KG-720038, Kyrgyzstan; 7Department of Biochemistry, Faculty of Veterinary Medicine, Afyon Kocatepe University, Afyonkarahisar 03200, Turkey; 8Department of Infectious Diseases and Preventive Medicine, Faculty of Veterinary Medicine, University of Life Sciences “King Mihai I” from Timisoara, 300645 Timişoara, Romania; 9Department of Biological and Environmental Sciences, Faculty of Nature and Life Sciences, University of Bejaia, Bejaia 06000, Algeria

**Keywords:** Algeria, *Mycobacterium bovis*, tuberculosis, ruminants, slaughterhouses

## Abstract

Tuberculosis (TB) is considered one of the most widespread and devastating zoonotic diseases in low-income countries, with a cosmopolitan distribution. The aim of this 10-year retrospective survey (from 2011 to 2020) was to determine the frequency of bovine, ovine, and goat tuberculosis in different local slaughterhouses across Constantine Province, Algeria. The control of livestock carcasses was systematically performed by veterinarian inspectors, after each stage of the slaughter process. The routine abattoir inspection included the detection of visible abnormalities on different organs and lymph nodes. The overall prevalence of tuberculosis recorded in slaughtered animals was 0.83%, with the following distribution among species: 2.73% in cattle, 0.001% in sheep, and 0.0% in goats. During the study period, there was a strong correlation (*R* = 0.82) (*p* < 0.01) between tuberculosis occurrence and the number of slaughtered cattle. Fluctuations in monthly TB prevalence ranged from 2% to 24.8% between 2018 and 2020, although there were no statistically significant correlations between infection and the age or gender of the animals, except for the year 2020 when a significantly higher (*p* = 0.017) percentage of TB cases were recorded in female cattle compared to male cattle. The average monthly weight of the confiscated livers and lungs ranged significantly (*p* ≤ 0.05) from 150 kg to 350 kg. The study results provide baseline data regarding livestock tuberculosis monitoring in the area of Constantine, Algeria, indicating that the disease incidence is not highly alarming, yet remains a serious public and animal health issue in the screened region.

## 1. Introduction

Infectious diseases are major public and animal health issues in both global hemispheres. Tuberculosis (TB) is a zoonotic, chronic, and debilitating contagious disease, which is a perfect illustration of a pathogen with no host or geographic boundaries, caused by members of the *Mycobacterium tuberculosis* complex [1]. *Mycobacterium tuberculosis* and *Mycobacterium bovis* are the main etiological agents of TB in a range of domestic and wild warm-blooded animal hosts. They are the main causative agents of pulmonary and extrapulmonary human tuberculosis, respectively [2,3]. Mycobacteria are generally aerobic microorganisms. However, anaerobic strains or species can occasionally occur, which are able to grow with minimal nutrients through a slow growth rate [1,2,3]. They belong to the Actinomycetales order, the Mycobacteriaceae family, and the *Mycobacterium* genus. They are acid-fast bacilli that are extremely robust in the environment due to their particular structural wall, which is rich in mycolic acids [4]. The primary route of transmission of *M. bovis* to humans and animals is indirect, most commonly occurring through the consumption of contaminated milk or other dairy products, which have not been subject to heat treatment [5]. It was noted that this bacterium can be transmitted, although less frequently, through the consumption of raw or undercooked contaminated meat [6]. Moreover, the direct airborne transmission of *M. bovis* has been reported via aerosol discharges, contact with sputum, saliva, urine, and manure, and watering or feeding sites [7]. Lymph nodes are the primary site of infection, followed by other organs, such as the lungs, liver, intestines, and kidneys, which may also be affected and progressively develop specific granulomatous lesions [8]. The size, color, and consistency of the lesions vary widely according to the stage of infection. Lesion sizes are microscopic or large enough to involve the greater part of, or the whole organ or tissue. The consistency ranges from caseopurulent, fibrocaseous fibro-calcified to calcified, although they may also be thin-walled purulent cavities Histopathological features of a granuloma show a central area of caseous necrosis with or without calcification, surrounded by macrophages, lymphocytes, plasma cells, neutrophils, epithelioid cells, and Langhans giant cells, while enclosed partially or completely by a fibrous capsule [9].

In domestic animals, bovine tuberculosis (bTB) has a significant economic impact and compromises the livelihood of populations. Tuberculous animals engender reduced milk and meat production, and significant losses of meat and offal. In addition, morbidity and mortality rates increased considerably, giving rise to additional treatment costs [10]. In low-income countries, TB is considered one of the most widespread and devastating zoonotic diseases, with a cosmopolitan distribution. The World Organization for Animal Health reported in 2019 that TB still occurs in 82 (44%) of the 188 countries and territories worldwide, of those that have declared their status for this disease [11]. The World Health Organization (WHO) estimates that bTB is involved in more than 3% of global human TB cases, localized mainly in low-income countries, particularly in Sub-Saharan Africa [12,13]. Bovine tuberculosis is an important animal health problem in the African continent. Statistical data reveals that there are reportedly 70,000 new cases of human TB through *Mycobacterium bovis* each year in the African continent [12,14]. Bovine tuberculosis is more widespread in countries with large cattle herds and where dairy production is an important industry [11]. The disease can infect a wide range of animals, including cattle, goats, buffalo, and wildlife. Considerable efforts have been made to control and eradicate bTB in Africa, including screening and slaughtering infected animals, vaccination programs, and improved management practices to reduce the risk of disease transmission. However, limited resources and infrastructure in many parts of the continent can make these efforts difficult. Bovine tuberculosis is also present in the Maghreb region with varying prevalence rates across countries and regions, including Morocco, Algeria, Tunisia, Libya, and Mauritania [13]. Factors that may be contributing to the spread and prevalence of the disease in these regions include poor biosecurity measures, inadequate testing and surveillance programs, and transboundary animal movements [12].

Algeria is regarded as a moderately endemic country, where tuberculosis is a major zoonotic disease that is prevalent at the countrywide level [15]. In response to this major risk, the Algerian health authorities have adopted a national eradication policy based on structured and rigorous sanitary prophylaxis. Since 1995, a multi-year cattle surveillance program has been implemented, and tuberculosis has been established as a mandatory declarable disease. In general, postmortem examinations for suspicious lesions at slaughterhouses are the official diagnostic procedures for tuberculosis in a lot of countries. In Algeria, the livestock, composed mainly of sheep, cattle, goats, and camels are screened yearly on an antemortem basis via a tuberculin skin test, or through a postmortem inspection of the meat in the slaughterhouses [16,17]. Veterinary inspectors perform livestock controls and play an important role in the sanitary surveillance of pulmonary infections [10]. It is known that slaughterhouse surveillance is regarded as a simple, cost-effective, and fairly reliable method of detecting animals with tuberculous, with a high level of sensitivity [18]. Therefore, slaughterhouses could be an important epidemiological starting point for the tracking of zoonotic TB [16]. In Algeria, especially in the eastern region, there are a few reports on the prevalence of animal tuberculosis, based on records from slaughterhouses [8,16,19,20]. Today, the real prevalence of ruminant tuberculosis is more or less worrying and the epidemiological situation is still incomplete in Algeria. It is very necessary to continue the epidemiological investigation on the magnitude of occurrence of this pathology in animal livestock, in order to control and eradicate this disease. Thus, the objective of the present retrospective 10-year survey was to determine the frequency of tuberculosis in ruminants (cattle, sheep, and goats), at different local slaughterhouses in the Constantine Province, from 2011 to 2020, in order to provide a general overview for the veterinarians, public health authorities, scientists, and the general public.

## 2. Materials and Methods

### 2.1. Study Area

The research was conducted in the Province of Constantine (36°16′ N, 6°37′ E), which is located in northeastern Algeria and has a land area of 231.6 km^2^ (Figure 1).

The landscape of this area is characterized by a mountain range (Tellian Atlas) and high plains. The monthly rainfall average is 525 mm, with minimum and maximum temperatures of −5.4 °C and 39.7 °C. The Province of Constantine is a varied agroecological region, with livestock made up of 39,454 cattle, 196,235 sheep, and 10,771 goats [21].

### 2.2. Slaughterhouse Postmortem Inspection Procedure

Data were obtained from different municipal slaughterhouses, supervised by the Provincial Veterinary Inspection of Constantine Province, from January 2011 to December 2020. In the enrolled slaughtering units, the control of livestock carcasses was regularly and systematically performed by a veterinarian inspector after each stage of the slaughtering process. The routine slaughterhouse inspection, as previously described, included the detection of visible abnormalities, meaning tubercular lesions, on different organs (e.g., lungs, liver, kidneys, uterus, spleen, udder, intestines, etc.), and lymph nodes [10]. In addition, incisions were performed in the trachea, bronchial, mediastinal, apical, medial retropharyngeal, submaxillary, mesenteric, hepatic, inguinal, and supramammary lymph nodes. The main characteristic of a tubercle lesion in ruminants consisted of a yellowish-white or grayish-white granuloma, enclosed in a capsule of varying thickness, often with caseous, caseocalcareous, or calcified contents [22]. The lymph nodes were sliced into multiple thin sections using knives and they were examined visually, under a bright light source, for the presence of TB-like lesions.

### 2.3. Prevalence Determination

The overall disease prevalence of the three animal species (cattle, sheep, and goats) was calculated from the available data, collected over a ten-year period (2011–2020). The number of slaughtered animals infected with tuberculosis was reported monthly and annually. The annual TB prevalence (%) rate was computed as the number of animals with suspect TB lesions divided by the number of animals examined postmortem, during that particular year. The seasonal prevalence (%) was also determined by calculating the total number of animals with TB lesions, recorded during all four seasons (spring, summer, autumn, and winter), divided by the total number of animals slaughtered, and examined for each season.

### 2.4. Statistical Analysis

The Microsoft Excel 2019 program was used to input and calculate all the recorded data. SPSS 25.0 software for Windows (SPSS Inc., Chicago, IL, USA) was used to perform the statistical analyses. Depending on the registered data, some of the results were described using descriptive statistics (means and standard deviations). When appropriate, the results were statistically interpreted by an independent sample t-test with a 95% confidence interval, or by an analysis of variance (ANOVA 1 parametric test) and a post hoc Tukey test (*p* ≤ 0.05). The comparison between the recorded epidemiological factors (e.g., age and sex) was performed by the chi-square (χ2) test. The non-parametric correlation between the annual or monthly prevalence rate and the number of animals slaughtered was assessed by Spearman’s test.

## 3. Results

Ten-year (2011–2020) slaughter statistics of the monitored ruminants (cattle, sheep, and goats) with tuberculosis lesions in Constantine Province, Algeria are summarized in Table 1. The large number of animals slaughtered during the study period—146,143 cattle, 327,290 sheep, and 6030 goats indicates the intense activity of these slaughterhouses.

The macroscopic lesions observed on tuberculous bovine carcasses were detected in the encephalon, lung, udder, liver, and chest cavity (Figure 2). There were significant differences (*p* ≤ 0.05) in the mean numbers of animals slaughtered per year from 2011 to 2020. Based on macroscopic postmortem screenings (Figure 2), the overall prevalence of tuberculosis recorded in slaughtered ruminants (cattle, sheep, and goats) was 0.83%, meaning a total of 3990 tuberculosis cases.

Figure 3 illustrates the recorded changes in the annual prevalence of tuberculosis in the monitored ruminants during the investigated period. The annual prevalence of tuberculosis was very low (0.01%), and statistically identical (*p* ≤ 0.05) in sheep. In goats, no cases of tuberculosis were recorded during the study period. On the other hand, the annual tuberculosis prevalence pattern among cattle remained statistically identical (*p* ≥ 0.05), with slight fluctuations from 1.09% to 2.34% between the years 2011 and 2018, followed by a significant increase (*p* ≤ 0.05), peaking at 6.39% in 2020. The correlation between the annual occurrence of tuberculosis and the number of slaughtered cattle was carried out using Spearman’s test. The two analyzed parameters were strongly correlated (R = 0.82) (*p* ≤ 0.01), during the study period (2011–2020).

Fluctuations in the monthly TB prevalences from 2% to 24.82% have been particularly observed during the time period from 2018 to 2020, as presented in Figure 4. The Spearman’s test showed a weak relationship (R = 0.36) (*p* ≤ 0.01) between the number of slaughtered cattle and the recorded monthly prevalence.

The seasonal prevalence of bovine tuberculosis seems to be significantly higher in spring and summer (*p* ≤ 0.05).

Results of the relationships between the sex, age, and tuberculosis lesion of cattle are shown in Table 2. There were no significant differences (*p* ≥ 0.05) recorded in the infection prevalence rates between young animals (<5 years old) and older ones (>5 years old).

Likewise, no difference (*p* > 0.05) was recorded in terms of gender when studying animals affected by tuberculosis with a single exception represented by significantly (*p* = 0.017) higher percentages of tuberculosis cases in females recorded during 2020, compared to the infection rates in males.

Figure 5 presents the annual progression of confiscated organs with tuberculous lesions in slaughterhouses from the Constantine Province, between 2011 and 2020. In the period 2011–2017, an average of 1000 kg of tuberculous organs (liver and lungs) was confiscated per year. In contrast, the total weight of confiscated organs reached 6000 kg in 2020 (*p* ≤ 0.05).

The average monthly weight of confiscated livers and lungs ranged significantly (*p* ≤ 0.05), from 150 kg to 350 kg, between 2018 and 2020 (Figure 6). 

## 4. Discussion

Tuberculosis is recognized as a major zoonotic disease with a worldwide distribution [11]. Bovine TB is a chronic infectious disease that primarily affects cattle, yet can also affect other domestic and wild animals, as well as humans. It is caused by the bacterium *Mycobacterium bovis* and is transmitted mainly by the inhalation of contaminated droplets or the consumption of unpasteurized dairy products [1,3]. It is ranked as the second or third disease after hydatidosis and fasciolosis, declared at the slaughterhouse level [18,23]. For this purpose, veterinary inspection is a crucial process in slaughterhouses, in order to minimize the transmission risk to humans through infected meat consumption. The Algerian government has put measures in place to control the spread of bovine tuberculosis, such as regular screening and the slaughtering of infected animals, as well as the use of pasteurization for milk, in order to reduce the risk of human infections. However, these efforts face challenges such as limited resources, lack of awareness among farmers and veterinarians, and difficulties in implementing effective surveillance and control measures in remote areas [16]. The meat inspection data seems to be a potential source of information in epidemiology surveys and plays an important role in preventive medicine. However, a limited number of studies have been conducted on the monitoring of tuberculosis in slaughterhouses in Algeria [8,16,19]. The present epidemiological study, extending over a 10-year period (2011–2021), was conducted to describe the epidemiological situation of tuberculosis in ruminants, based on the records from different local slaughterhouses in Constantine Province.

The present study revealed that the rate of sheep slaughtering was significantly higher than for cattle and goats (68.5%, 30.26%, and 1.24%, respectively). It should be highlighted that the demand for red meat in Algeria is mainly focused on sheep and beef, especially for weddings and religious parties (Eid Al-Adha and Ramadan). In Algeria, the Ministry of Agriculture and Rural Development reported in 2012 that the national herd consisted of 34 million ruminants, including 28.9 million sheep, 4.9 million goats, and 1.9 million cattle [24]. Furthermore, Algeria is the world’s fifth largest producer of ovine meat [25]. Depending on the region, food preferences and gastronomic habits also play a role in meat consumption. Sheep meat (60%) appears to be more popular than beef (40%) in eastern Algeria [26].

Based on the collected data, male cattle (80%) dominated the numbers among all slaughtered animals. This is explained by the fact that males are destined for fattening and meat production. However, females are used in bovine reproduction and milk production [1]. In addition, Algerian legislation strictly prohibits the slaughter of female cattle, except for medical reasons (e.g., infertility and trauma) [27]. It was also noted that the Algerian consumer would prefer the bovine meat of young animals (75%), because of its tenderness and due to the animal’s fattening degree. This finding is similar to the results reported previously by Hamiroune et al. [19] and Mimoune et al. [23].

The Algerian health authorities deployed physical and personal resources to eradicate tuberculosis, declaring it a notifiable contagious disease. However, this disease remains moderately endemic, with regional disparities [16]. Based on a detailed postmortem inspection, our results show that the prevalence of bTB is low in cattle and negligible in sheep (2.73% and 0.01%, respectively). The statistics are certainly underestimated since the national strategy focuses mainly on screening for bTB through tuberculization testing and postmortem inspection of meat. The molecular or immunological diagnosis would require exorbitant costs [28]. The rate of bovine tuberculosis calculated in our study was nearly similar to that recorded previously in the Bejaia area [16]. However, many studies in Algeria reported higher cattle bTB prevalence rates than in our data [1,5,8,26]. Likewise, the prevalence rate reported in the present study was lower than those obtained in previous studies conducted in Morocco (3.7% and 4.6%) [29,30], and Tunisia (3.39%) [31]. High levels of bTB have also been reported in other African countries, including Ghana (5%) [32], Cameroon (11.22%) [33], and Ethiopia (4.7%) [9]. In Turkey, the prevalence of tuberculosis lesions was found to be 7.7% in cattle after macroscopic examinations [34]. Another study reported that out of the 100,196 examined animal carcasses, 5221 (prevalence of 5.21%) showed tuberculous gross lesions in Sicily, southern Italy [35]. On the other hand, this prevalence is higher than the rates reported based on postmortem slaughterhouse surveillance in Burkina Faso (1.07%) [36], and Zambia (1.56%) [37]. The apparent prevalence of bovine tuberculosis in the total number of animals and in herds slaughtered in 2009 in the State of Mato Grosso (Brazil) was 0.007% and 0.61%, respectively [38]. In Romania, the prevalence of bovine tuberculosis postmortem found in bovine organs showing the lesions in the lung and carcass were 3% and 1.5%, respectively [39]. The tuberculosis prevalence rate among slaughtered sheep in the present study (0.01%) is lower compared to the previous surveys that reported prevalence in Algeria by Sahraoui et al. [5], and in Cameroon by Awah-Ndukum et al. [40] (6.03% and 2.30%, respectively). Previous studies reported an ovine tuberculosis prevalence of 15% in Ethiopia [41], and 32% in Spain [42]. These variations might be attributed to many factors, such as animal origin, age, animal husbandry practices, and hereditary resistance. This could also be due to inspection conditions and different control practices. It is known that sheep can be infected with *M. bovis* and used as hosts if the infection level is relatively high. Maintenance of the infection is assured in the sheep population by the presence of infected animals or by a wild reservoir [41]. The low prevalence of bTB in sheep and goats could be attributed to their lack of contact with other animal species. Our results agree with reports conducted by several researchers that tuberculosis is rare in sheep owing to their breeding conditions [43,44]. The prevalence of TB is different in various species due to malnutrition, pregnancy, and concurrent infection that may affect the immune responsiveness of animals.

The specific survey of slaughtered cattle, which extended from 2018 to 2020, has reported significant deductions. In this survey, the seasonal prevalence in cattle (seen from 2018 to 2020) appears to have a consistent tendency across the seasons in Constantine Province, with a significant peak during the dry season, notably in spring or summer. This pattern of bTB incidence corroborates those of the monthly ones (Figure 3), while the respective peaks of 24.82% and 11% are reached in the same time period, extending from June to September. This outcome must only be consistent with the significant consumer demand throughout the holiday seasons and during family gatherings, which take place during the spring and summer seasons. Our results concurred with earlier findings from several Algerian localities [1,8,26]. Since bacillus has a long incubation period, the influence of seasonality, as a risk factor, is often debatable since it might delay the specific lesions of bTB at any time of the year [30,45]. The prevalence of tuberculosis may be influenced by herd movements during the winter season, which reduces the risk of bTB transmission between animals. However, the herds may be affected by bTB during the grazing season (e.g., summer season) when they come in close contact with other animals. In addition, the *Mycobacterium* spp. seems resistant in pastures and in water environments [12,46,47].

The age and gender of animals have been reported to be risk factors for tuberculosis [48]. In the present study, the outcomes of the assessment of risk factors related to the gender and age of the slaughtered cattle, within the three focus years, seem to be fairly conclusive (Table 2), whereby age and gender are limited factors in terms of favoring bTB contamination, although age-wise, the affected categories lean towards adulthood. As for the influence of animal gender, females seem to be more susceptible. Similar findings have been reported by several authors [1,8,16,26]. In another study, the prevalence of diagnosed tuberculosis in slaughterhouses was higher in females than in males (16.3 vs. 7.3 cases/100,000, respectively) [49]. Generally, female cattle are not necessarily more susceptible to bovine tuberculosis than male cattle. However, some factors might contribute to a higher incidence of bovine TB in females. It is known that females have a longer life span. In addition, females are usually bred longer than male cattle because of their reproductive activity and milk production. Furthermore, cows might go through periods of immunosuppression during lactation, leading to an increased susceptibility to infection. Moreover, females are typically more sociable than males and can interact more easily with other animals, raising the risk of disease transmission [50]. However, Borham et al. [48] and Ullah et al. [51] reported that there was a significantly higher risk for males to develop bTB lesions. It has been equally shown that the frequency of bTB in older cattle is more common [52]. This could be explained by the slow and chronic characteristics of tuberculosis [13,48]. In addition, the presence of TB lesions in younger bovines would probably be linked to *M. bovis* transmission in maternal milk [13,48]. However, further studies based on molecular tools are still necessary to clarify this hypothesis.

Bovine tuberculosis is characterized by typical lesions called tubercles that contain *Mycobacterium* and are localized in different tissues, such as the lung, lymph nodes, and intestines [48]. Many investigations recorded that lung condemnation rates were important due to the tuberculosis lesions in slaughtered animals [53,54,55]. It is also reported that histopathology examinations of caseous tuberculosis were observed in 57% of the lungs, 57% of the bronchial, and 53% of the mediastinal lymph nodes [34]. Nevertheless, the lungs are considered to be the primary site of bTB infection because it is the first point of exposure to inhaled bacteria. *M. bovis* has a large alveolar surface area for bacterial colonization and provides an ideal environment for bacterial growth, and survival, with high oxygen content, especially for the establishment of an infection. In addition, the lungs are rich in macrophages, which play a crucial role in the immune defense against bacterial infections [56,57,58]. From the results of the 10-year epidemiological survey, the data obtained through the postmortem inspection revealed significant losses of meat and offal, while the bTB predilection site was the lung. This finding agrees with several studies, which focused on the incidence of TB and its locations in animals from slaughterhouses [1,8].

## 5. Conclusions

The findings of the present study provide baseline data in terms of ruminant tuberculosis monitoring in Constantine Province, Algeria. The obtained results showed that the infection prevalence rates are not very alarming, although this disease remains a serious problem for public and animal health. However, other approaches such as bacteriological and histopathological examinations, are necessary to further strengthen these results. Further, in order to develop efficient control programs, additional investigations are required based on (i) the molecular analysis of *Mycobacterium* strains from animals, (ii) their genetic diversity, and (iii) molecular markers of drug resistance. In addition, investigations on the determination of risk factors in livestock and the estimation of the recorded economic losses are recommended. 

## Figures and Tables

**Figure 1 life-13-00817-f001:**
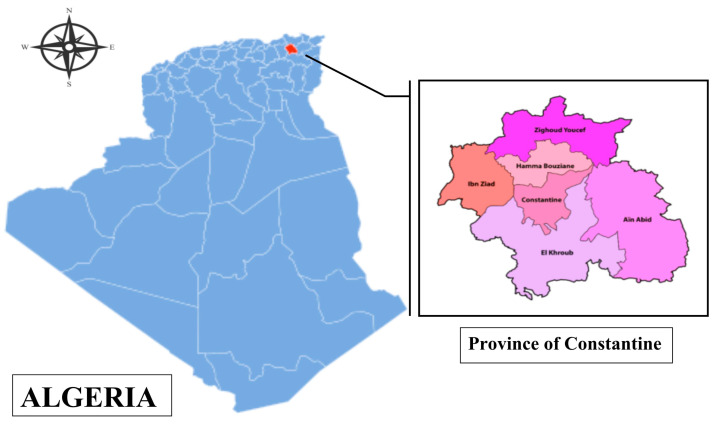
Map of Constantine Province, Algeria.

**Figure 2 life-13-00817-f002:**
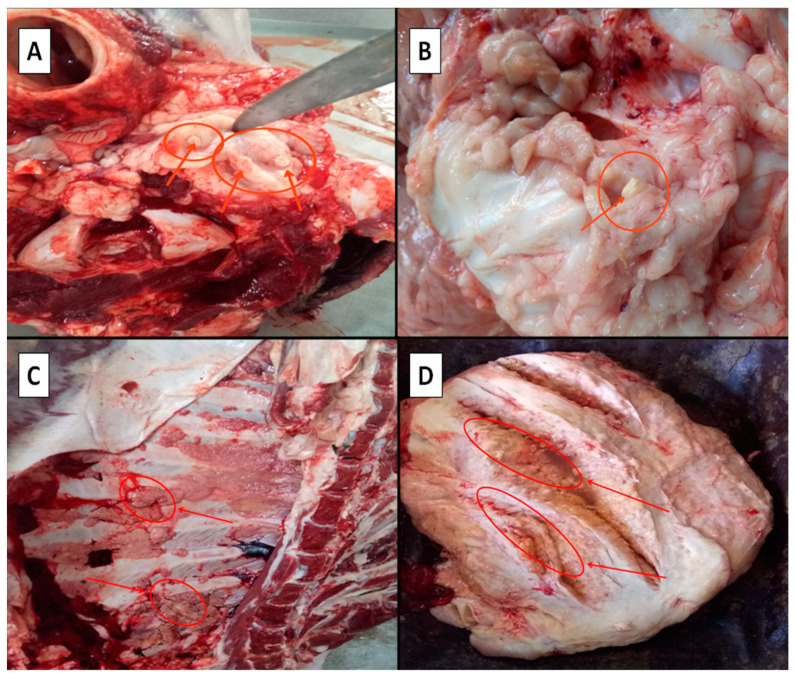
Tuberculous lesions indicated by red arrows on the encephalon (**A**), lung (**B**), udder (**C**), and chest cavity (**D**) were observed in abattoirs of Constantine Province, Algeria.

**Figure 3 life-13-00817-f003:**
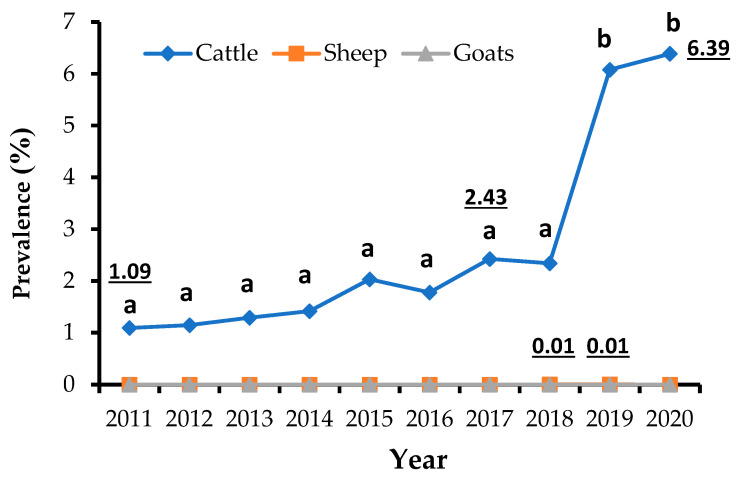
Annual trends of animals infected with tuberculosis in slaughterhouses, from 2011 to 2020, in Constantine Province, Algeria. Different letters on the same curve indicate a statistically significant difference (Tukey’s test, *p* < 0.05).

**Figure 4 life-13-00817-f004:**
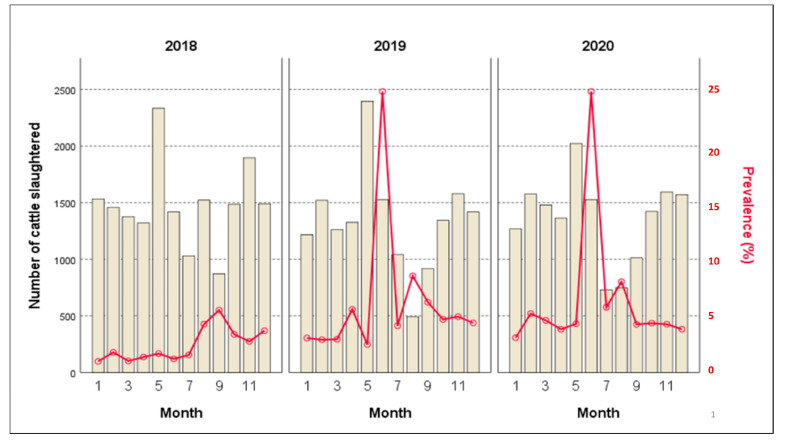
Correlation between the mean number of slaughtered cattle and the monthly prevalence rates during the period 2018–2020 in the Constantine Province, Algeria.

**Figure 5 life-13-00817-f005:**
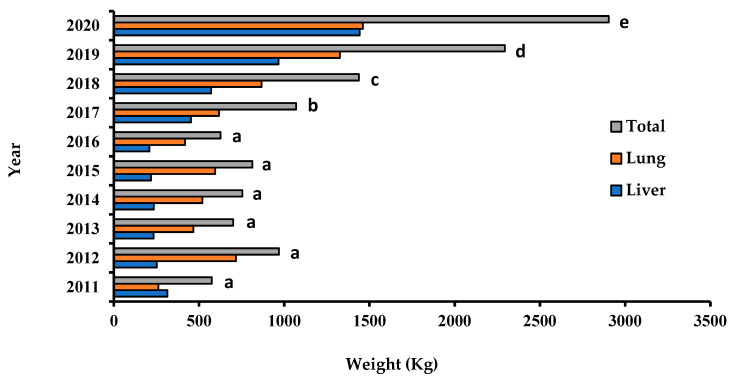
Annual weight of bovine livers and lungs with tuberculosis lesions confiscated during the period 2011–2020 in slaughterhouses from the Province of Constantine. Different letters on the same bar indicate a statistically significant difference (Tukey’s test, *p* < 0.05).

**Figure 6 life-13-00817-f006:**
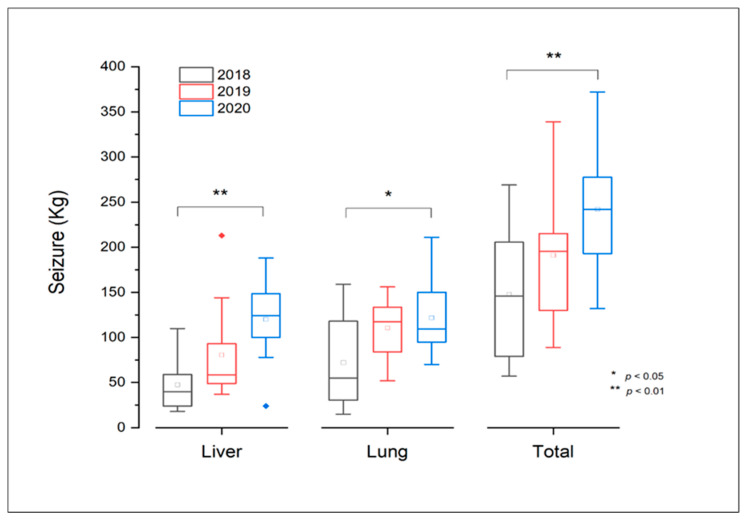
Statistical data regarding confiscated bovine livers and lungs affected by tuberculosis during the period 2018–2020 in slaughterhouses from Constantine Province, Algeria.

**Table 1 life-13-00817-t001:** Tuberculosis rates of infected animals between 2011 and 2020 in Constantine Province.

Slaughtered Species	Cattle	Sheep	Goats	Total of Animals
Slaughtered number	146,143	327,290	6030	479,463
Mean ± SD	14,614.30 ± 1636.31 ^a^	33,089.30 ± 8119.91 ^b^	597.40 ± 453.01 ^c^	48,301 ± 9445.25 ^d^
Min-Max	12,914–17,733	22,340–44,704	6–1411	36,128–62,430
Slaughtered Percentage (%)	30.26 ^a^	68.50 ^b^	1.24 ^c^	100
Number with tuberculosis lesions	3986	4	0	3990
Global Prevalence (%)	2.73 ^a^	0.001 ^b^	0 ^b^	0.83

Values with a different superscript letter in the same row are significantly different (*p* < 0.05).

**Table 2 life-13-00817-t002:** Prevalence rates (%) of tuberculosis in slaughtered infected animals depending on age and sex between 2018 and 2020 in Constantine Province, Algeria.

Year	Age and Sex	Number of Slaughtered Animals	Number of Infected Animals	Prevalence (%)	95% CI	*p*
2018	<5 Years	12,833	311	2.42	2.16–2.69	0.204
	>5 Years	4900	104	2.12	1.71–2.52	
	Male	13,655	332	2.43	2.17–2.69	0.142
	Female	4078	83	2.04	1.60–2.47	
2019	<5 Years	12,288	720	5.86	5.44–6.27	0.223
	>5 Years	3751	240	6.40	5.62–7.18	
	Male	13,068	768	5.88	5.47–6.28	0.224
	Female	2971	192	6.46	5.58–7.35	
2020	<5 Years	12,481	782	6.27	5.84–6.69	0.230
	>5 Years	3834	261	6.79	6.00–7.59	
	Male	13,499	835	6.18	5.77–6.58	0.017
	Female	2816	208	7.39	6.43–8.37	

## Data Availability

Data is contained within the article.

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
