# Peer review of "Estimated Prevalence of Tuberculosis in Ruminants from Slaughterhouses in Constantine Province (Northeastern Algeria): A 10-Year Retrospective Survey (2011–2020)"

_life, 2023, doi:10.3390/life13030817_

Round 1

Reviewer 1 Report

Dear authors,

Please find below some tips to improve your article, I think in general it must be a little improved in terms of the global impact of your research.

 Kind regards. 

Introduction

Line- 70 – There are anaerobic mycobacteria, please revise.

https://pubmed.ncbi.nlm.nih.gov/28874407/

Material and methods

2.4. Statistical analysis – Please inform us if the data had normal distribution, and which tests and conversions you performed prior ANOVA and Tukey test. Counting and percentages data are generally known for non-normal distribution.

Discussion

For a global audience, I think the discussion section must add a paragraph relating the findings of this research to what happens in other parts of the globe, and how these data can contribute to the science regarding TB prevention. 

Author Response

Reviewer#1

Dear authors,

Please find below some tips to improve your article, I think in general it must be a little improved in terms of the global impact of your research.

Kind regards.

Dear reviewer, our sincere thanks for taking the time to review this manuscript, and your close attention to detail. We highly appreciate your overall positive feed-back regarding the quality of the manuscript! Please see below for our responses to your comments:

Comment 1: Introduction, Line- 70 – There are anaerobic mycobacteria, please revise https://pubmed.ncbi.nlm.nih.gov/28874407/

We have well checked it. It was an error. Thank you for this remark.

Comment 2: Material and methods, 2.4. Statistical analysis, – Please inform us if the data had normal distribution, and which tests and conversions you performed prior ANOVA and Tukey test. Counting and percentages data are generally known for non-normal distribution.

Thank you very much for this relevance. You are absolutely right.

In our case, we conducted a retrospective study on recorded data from veterinary services. Therefore, our analytical strategy is totally impacted by the nature of the recorded data. Initially, we tested the homogeneity of the data before adopting the appropriate parametric or non-parametric tests. When permissible, we were able to perform descriptive analysis, analysis of variance and even post hoc examination. Otherwise, for the non-parametric tests, they do not make any assumptions about the type of distribution of the data. They are based solely on the numerical properties of the samples. The paragraph of the “statistical analysis” has been changed and corrected.

Comment 3: Discussion, For a global audience, I think the discussion section must add a paragraph relating the findings of this research to what happens in other parts of the globe, and how these data can contribute to the science regarding TB prevention. 

We agree with you. In the revised version, we added some recent references about bTB post-mortem inspection.

Thank you again!

Reviewer 2 Report

The novelty of this study is not clear, please add the novelty immediately before the objective of this study, moreover, the introduction is a little long. More information has to be added concerning the number of farms that were used to collect the data.

in the simple abstract, you don't need to insert P values in it, it is just a simple abstract.

Line 35, R in place of "Rho"

Line 52, it seems a missing article here e.g., < or =, another issue is the Rho to be "R"

The introduction is a little long, may you can be concentrated on the description of tuberculosis status in Algeria.

Line 144, for which animal? their breeds? if they are females or what?.

Line 286, a suitable reference can be added here.

Line 305, wiled reservoir!!

Author Response

Reviewer#2

Dear reviewer, our sincere thanks for taking the time to review this manuscript, and your close attention to detail. We highly appreciate your overall positive feed-back regarding the quality of the manuscript! Please see below for our responses to your comments:

Comment 1: The novelty of this study is not clear, please add the novelty immediately before the objective of this study, moreover, the introduction is a little long. More information has to be added concerning the number of farms that were used to collect the data.

We have added the mean motivation of the present study in the section "Introduction". Please, see the revised version. We think it is clearer now. In fact, we have not the data detailed on farms. All slaughtered animals in abattoir are mixed in the data collected.

Comment 2: In the simple abstract, you don't need to insert P values in it, it is just a simple abstract.

It is noted. We modified in the revised version.

Comment 3: Line 35, R in place of "Rho"

This is corrected in the revised version

Comment 4: Line 52, it seems a missing article here e.g., < or =, another issue is the Rho to be "R"

This is corrected in the revised version

Comment 5: The introduction is a little long, may you can be concentrated on the description of tuberculosis status in Algeria.

We completed in the revised version some information about tuberculosis status in Algeria.

Comment 6: Line 144, for which animal? Their breeds ? If they are females or what?.

TB data were analyzed using the following factors: sex (male and female), age (<5 years and >5 years). The authors regrettable inform the reviewer that data about the "breed" are not available. It is mentioned in section "Statistical analysis".

Comment 7: Line 286, a suitable reference can be added here.

This is an observation from the Algerian field. We want to say that the data on animal tuberculosis are not taken with an alarming eye, despite of the efforts made to eradicate the pathology. For this reason, there is not a reference.

Comment 8: Line 305, wiled reservoir!!

It is correct. We mean “wild reservoir”

Thank you again!

Reviewer 3 Report

Dear authors, in general I think the article presented is interesting. The subject matter is of great relevance and the information gathered could be useful to the scientific community.

I report below some general comments and specific observations to be addressed to improve the manuscript.

Proofreading of the draft for possible typos.

Keyword: I would add tuberculosis and eliminate prevalence.

Introduction: for completeness of information, it would be helpful to expand the description (line 83) on the characteristic lesions of the disease as in lines 348-355.

Materials and Methods: it would be necessary to specify in detail how the data collection took place. Specifically: is there a registry for inspectors? Were all carcasses slaughtered in all slaughterhouses considered or only a sample? Did the animals come exclusively from the area under investigation? Was the diagnosis of suspicion followed by laboratory confirmation?

Discussion and Conclusions: the limitations of the study should be specified.

With the hope that my comments will be helpful

Regards,
the reviewer

Author Response

Reviewer#3

Dear reviewer, our sincere thanks for taking the time to review this manuscript, and your close attention to detail. We highly appreciate your overall positive feed-back regarding the quality of the manuscript! Please see below for our responses to your comments:

Comment 1: Keyword, I would add tuberculosis and eliminate prevalence.

We deleted “prevalence” and added “tuberculosis”. Please, see the revised version.

Comment 2: Introduction, for completeness of information, it would be helpful to expand the description (line 83) on the characteristic lesions of the disease as in lines 348-355.

It is a pertinent remark. We added some information on the characteristic tuberculosis lesions. We hope that this will facilitate the understanding.

Comment 3: Materials and Methods: it would be necessary to specify in detail how the data collection took place. Specifically: is there a registry for inspectors? Were all carcasses slaughtered in all slaughterhouses considered or only a sample? Did the animals come exclusively from the area under investigation? Was the diagnosis of suspicion followed by laboratory confirmation?

Each slaughter, the animals are recorded per day and the diagnosed case of tuberculosis in the registry. After then, the data are entered and stored in electronic file. And, all carcass slaughtered (also tuberculosis cases) are considered in the present survey. Once the carcass is diagnosed with tuberculosis, this is this one is seized. The diagnosis of suspicion is not followed by laboratory confirmation.

Comment 4: Discussion and Conclusions, the limitations of the study should be specified.

In the revised version, we improved this part.

Thank you again!

Round 2

Reviewer 1 Report

Dear authors,

Thank you for revising according.

Just revise it.

Line 60-61 - Mycobacteria are anaerobic or microaerophilic microorganisms, with a slow growth rate.

Mycobacteria is generally aerobic, however anaerobic strains or species occur. In the first review I remember it sounded as only aerobic existed.

The rest of the article if fine for me.

Kind regards.

Author Response

Reviewer#1

Dear authors,

Thank you for revising according.

Just revise it.

Dear reviewer, our sincere thanks for taking the time to review this manuscript, and your close attention to detail. We highly appreciate your overall positive feed-back regarding the quality of the manuscript! Please see below for our responses to your comments:

Comment: Line 60-61 - Mycobacteria are anaerobic or microaerophilic microorganisms, with a slow growth rate. Mycobacteria is generally aerobic, however anaerobic strains or species occur. In the first review I remember it sounded as only aerobic existed.

Answer: We have rephrased the sentence resulting in: “Mycobacteria are generally aerobic microorganisms. However, anaerobic strains or species can occasionally occur, able of growing with minimal nutrients with a slow growth rate [1-3].”

The rest of the article if fine for me.

Kind regards.

Thank you again!

Reviewer 2 Report

The manuscript is improved and I'm satisfied with the corrections, it can be accepted for publication.

Author Response

Dear reviewer, our sincere thanks for taking the time to review this manuscript. We highly appreciate your overall positive feed-back regarding the quality of our submission!

Yours faithfully,

Dr. Imre,

On behalf of the research team